The effect of endoscopic polidocanol carbon dioxide foam for internal hemorrhoids: a retrospective study

Xiang Jiahui 1
Li Shichao 1
Yu Tengjiang 1
Jiang Qingfeng 1
Jiang Xia 2
Lan Yong 2 1025213332@qq.com
1 The Anorectal Department, The Affiliated Traditional Chinese Medicine Hospital, Southwest Medical University , Luzhou, Sichuan , China
2 The Civity Mirror Department, The Affiliated Traditional Chinese Medicine Hospital, Southwest Medical University , Luzhou, Sichuan , China
Anson Lesley
Electronic publication date: 2025 Nov 17
Publication date: 2025
Volume: 13
Electronic Location ID: e20252
Received 2024 Oct 31; Accepted 2025 Sep 25
Copyright: © 2025 Xiang et al.
Copyright year: 2025
Copyright holder: Xiang et al.
License: This is an open access article distributed under the terms of the Creative Commons Attribution License, which permits unrestricted use, distribution, reproduction and adaptation in any medium and for any purpose provided that it is properly attributed. For attribution, the original author(s), title, publication source (PeerJ) and either DOI or URL of the article must be cited.
License URL: https://creativecommons.org/licenses/by/4.0/

Keywords: Hemorrhoids, Endoscopic sclerotherapy, Polidocanol, Carbon dioxide, Air

Funding: Sichuan Provincial Administration of Traditional Chinese Medicine Special Project on Traditional Chinese Medicine 2021MS360 Southwest Medical University 2021ZKQN141, 2022QN082 This work was supported by the Sichuan Provincial Administration of Traditional Chinese Medicine Special Project on Traditional Chinese Medicine (No. 2021MS360), the Southwest Medical University school-level scientific research project (2021ZKQN141, 2022QN082). The funders had no role in study design, data collection and analysis, decision to publish, or preparation of the manuscript.

==============================
Aim

This article aims to explore the effectiveness of polidocanol carbon dioxide foam as a therapy for internal hemorrhoids.

Methods

A retrospective analysis was conducted on 158 patients who received endoscopic polidocanol foam sclerotherapy for internal hemorrhoids between October 2022 and September 2023. Among the patients, 78 underwent endoscopic polidocanol sclerotherapy with polidocanol air foam (control group), while 80 received polidocanol carbon dioxide foam (study group). The primary outcomes of this study were clinical effect and the incidence of complications 1 month after surgery. Patient self-reports of anal bleeding and anal prolapse served as the foundation for establishing three categories of clinical efficacy criteria. The clinical effect was determined based on the efficacy index, which was employed to assess treatment effectiveness. The occurrence of several postoperative complications was documented, specifically including anal pain, anal swelling, urinary retention, perianal infection, ectopic embolization, and anal edema. Secondary outcomes encompassed the surgical cost, duration of hospital stay, and the administered dosage of polidocanol.

Result

The treatment success at the end of the sclerotherapy session in the study and control group were 85.0% and 83.3%, respectively (P = 0.829). One month following endoscopic sclerotherapy, the study group exhibited an effective rate of 98.8%, while the control group showed a rate of 98.0%. Compared with the clinical effects one month after surgery of the two groups of patients, postoperative complications, surgical cost, hospitalization time and the dosage of polidocanol, we found 11 patients with anal pain and 12 patients with anal swelling in the control group. Within the study group, three patients experienced anal pain, while four patients presented with anal swelling. The study group exhibited significantly fewer patients with anal pain and anal swelling compared to the control group (P < 0.05), and importantly, no severe complications were observed. Both groups exhibited comparable clinical effects, surgical costs, hospitalization durations, and dosages of polidocanol, with no statistically significant differences observed (all P > 0.05).

Conclusion

The clinical effectiveness of endoscopic polidocanol carbon dioxide foam is on par with that of polidocanol air foam. In sclerotherapy for internal hemorrhoids, the use of endoscopic polidocanol carbon dioxide foam can significantly decrease complications such as anal swelling and pain.

Introduction

Hemorrhoids are a common disease worldwide, with a peak prevalence between 45 and 65 years of age in both men and women (Xie et al., 2022). Epidemiological survey results in China show that the prevalence of anorectal diseases is as high as 50.1%, among which 98.08% of patients have hemorrhoid symptoms (Johanson & Sonnenberg, 1990). Patients often show repeated hematochezia, anal swelling and pain etc., which have severe effects on their health and life (Margetis, 2019).

Nowadays, endoscopic sclerotherapy is widely practised as a minimally invasive procedure to treat internal hemorrhoids and provides an alternative to conventional hemorrhoidectomy (Hachiro et al., 2011), it has simple operation, less trauma, good efficacy, low cost and fewer postoperative complications (Nastasa et al., 2015). Common sclerosants have aluminum potassium sulfate tannic acid (ALTA), phenol in almond oil (PAO) and polidocanol. Polidocanol has the advantages of being safer and more effective, and it can effectively reduce complications and improve patient satisfaction (Mishra et al., 2020).

But at present, most hospitals use air foam hardening agent to treat internal hemorrhoids, which has the characteristics of slow postoperative absorption, long duration of anal swelling and so on. A search of relevant articles found that with the development of sclerotherapy, the treatment of varicose veins was significantly reduced if CO2 rather than air was employed to make the sclerosing foam for chemical ablation of superficial veins of the lower extremity (Morrison et al., 2008). Also, in a literature report, patients with CO2/O2 foam were well tolerated with fewer similar therapeutic side effects than air foam (Morrison et al., 2010). In a study that specifically investigated the neurological side effects associated with migraine for the use of sclerotherapy and physiological foam, the use of CO2 or CO2/O2 foam should be considered for those patients at increased risk of neurological side effects, such as patients with migraine with aura and those with known Patent foramen ovale (Wong, 2015). The half-life of air and carbon dioxide binding with polidocanol is different. This leads to the different stability of the foam (Bai et al., 2020). The combination of air and polidocanol has the longest half-life, which results in a prolonged contact time between the sclerosant and the tissue. Furthermore, air is poorly absorbed by the human body, resulting in extended stimulation that can initiate a cascade of complications. Carbon dioxide, a prevalent physiological gas, often serves as the basis for creating pneumoperitoneum during laparoscopic surgeries. Its readiness to be absorbed by the human body may offer a solution to these problems. Although similar findings have been documented in studies examining the use of carbon dioxide foam in varicose vein treatments and other sclerotherapies, no detailed report has been published on the distinct effects of polidocanol foam sclerosing agents formulated with either air or carbon dioxide in the treatment of internal hemorrhoids. Hence, this article conducted a retrospective, comparative analysis to assess the effectiveness of endoscopic polidocanol carbon dioxide foam sclerotherapy vs polidocanol air foam sclerotherapy for treating internal hemorrhoids.

Materials and Methods

Study design and setting

The protocol was approved by the Medical Ethics Committee of the Affiliated Traditional Chinese Medicine Hospital to Southwest Medical University (approval number: BY2023033). The flowchart of this trial is shown in Fig. 1.

Figure 1 The flowchart of this trial.

A retrospective analysis was performed on 158 patients who underwent endoscopic polidocanol foam sclerotherapy for internal hemorrhoids in the Anorectal Department of the Affiliated Traditional Chinese Medicine Hospital to Southwest Medical University from October 2022 to September 2023. Among them, 78 cases were treated with polidocanol air foam (control group), and 80 cases were treated with polidocanol carbon dioxide foam (study group).

Participants

All the patients were from the Affiliated Traditional Chinese Medicine Hospital to Southwest Medical University.

Inclusion criteria

Participants were required to fulfill the following criteria: (1) They must be patients diagnosed with degree I–III internal hemorrhoids according to the widely accepted Goligher classification of internal hemorrhoids (Dekker et al., 2022). (2) They should be willing to undergo endoscopic sclerotherapy. (3) Their clinical data must be complete.

Exclusion criteria

Participants were excluded if they fulfilled any of the following criteria: (1) presence of other diseases, including incarcerated hemorrhoids, ring mixed hemorrhoids, perianal abscess, anal fissure, anal fistula, rectal prolapse, intestinal tumors, or other comorbidities; (2) prior treatment with hemorrhoid ligation; (3) existence of severe primary diseases affecting important organs such as the heart, brain, liver, kidney, or lungs; (4) abnormal coagulation function or current use of antiplatelet anticoagulants.

Withdrawal/termination criteria

Participants meeting the following criteria were to be withdrawn or terminated from the trial: (1) incomplete or missing data during follow-up, rendering the curative effect indeterminate; (2) receipt of traditional surgery or other treatments during the observation period; (3) failure to cooperate with follow-up observers.

Interventions

The control group patients gave informed consent before surgery, fasted, and underwent an enema on the procedure day. The left lateral decubitus position was adopted during the surgery, and a transparent cap was affixed to the tip of the colonoscope. After completing the ileocolon examination, internal hemorrhoids were detected, and a thorough evaluation of their precise location and severity was conducted. The foam hardening agent was prepared by the Tessari technique (Xu et al., 2016) (Fig. 2). A U-shaped inverted mirror or a transparent cap was employed to guarantee unobstructed vision throughout the surgical process. The injection needle was inserted at a 30° to 40° angle, aiming precisely at the injection sites situated in the submucosal vessels of the hemorrhoids, positioned just over 0.5 cm above the dentate line. A volume ranging from 1 to 2 ml of polidocanol sclerosing agent was administered at each injection site, with a total of 4 to 5 carefully chosen points for the treatment. The total injection volume typically did not exceed 20 ml. Specifically, the injection should be administered to achieve a grey and white appearance of the mucosa at the injection site, accompanied by distinctly visible blood vessel textures. The needle should be withdrawn slowly while the injection was being administered. Following the injection, it was recommended to apply pressure to the injection site for 5 to 8 s to minimize bleeding (Fig. 3). After surgery, patients were advised to follow a liquid diet, maintain regular bowel movements, and used laxatives as needed to avert constipation. Patients were followed up for one month. Those in the treatment group underwent an injection of 3% polidocanol carbon dioxide foam (Fig. 4). The preoperative preparations, injection site, method, depth, and volume were all kept consistent with the control group. The preparation of the foam hardening agent also employed the Tessari technique. The control group collected 8 ml of indoor air, whereas the treatment group gathered 8 ml of carbon dioxide. The subsequent treatment procedures were identical for both groups. Both groups received the same postoperative care. All surgeries were performed by attending physicians or higher-level medical professionals. The surgeons had undergone endoscopic trained at the same hospital, ensured their proficiency.

Figure 2 Preparation of polidocanol air foam hardener: (A) instrument preparation, (B–D) 2 ml of 3% polidocanol injection and 8 ml of indoor air are mixed rapidly for 15–20 times.

Figure 3 (A–D) The injection site for 5 to 8 s to prevent bleeding.

Figure 4 A total of 3% polidocanol carbon dioxide foam injection.

Outcomes

The primary outcome of this study was the clinical effect and incidence of complications one month post-surgery. Patients self-reported instances of anal bleeding and anal prolapse were used as the basis for evaluating clinical efficacy, categorized into three classes: (1) cure—very satisfied with no or mild symptoms; (2) improvement—occasional symptoms; and (3) invalid—no improvement or worsening of symptoms (Xie et al., 2022). In order to evaluate the clinical effect more accurately, we defined the clinical effect by the efficacy index (efficacy index (%) = [(pre-treatment score − post-treatment score)/pre-treatment score] × 100%). We defined anal bleeding and anal prolapse scores by the following method (anal bleeding score: 0 was < 1 episodes per month, 1 was <1 episode per week, 2 was 1–3 episodes per week and 3 was 4 more episodes per week; anal prolapse score: no anal prolapse = 0, mild anal prolapse = 1, moderate anal prolapse = 2, severe anal prolapse = 3) (Giamundo et al., 2018; Rørvik et al., 2019). We defined the efficacy index as follows: a value of ≥90% indicates a cure, 30% to 89% signifies improvement, and <30% was classified as invalid. By calculating the efficacy index for anal bleeding and anal prolapse, we determined that both cure and improvement qualify as effective outcomes. The incidence of various postoperative complications, including anal pain, anal swelling, urinary retention, perianal infection, ectopic embolization, and anal edema, was meticulously recorded. Secondary outcomes included the dosage of polidocanol, the cost of surgery, and the length of hospital stay.

Statistical analysis

Data were analyzed by using the SPSS 25.0 statistical software. Count data were represented by the number of cases and percentage (n (%)). Comparisons between groups were performed using the χ2 test. Rank data were compared by using the rank-sum test. Measurement data were expressed as mean ± standard deviation ( x¯±s). Comparisons between the groups were performed by using the t-test. P < 0.05 was considered as a statistically significant difference.

Result

A total of 168 patients consented to participate in the study. General data for both groups were presented in Table 1. The mean ages of the patients in the study and control groups were 48.76 ± 12.05 years and 50.71 ± 10.46 years, respectively. The study group consisted of 56 males and 24 females, whereas the control group included 60 males and 18 females. No significant differences were observed in general demographics, including age and sex, between the two groups (all P > 0.05).

Table 1 General data of patients in both groups.

Baseline date	Study group (n = 80)	Control group (n = 78)	χ2/t	P value	
Sex			0.970	0.325	
Male	56 (70.0%)	60 (76.9%)			
Female	24 (30.0%)	18 (23.1%)			
Age (year)	48.76 ± 12.05	50.71 ± 10.46	−1.081	0.294	
Height (cm)	167.69 ± 8.69	168.11 ± 6.59	−0.348	0.728	

Table 2 presented a comparison of the clinical effects between the study and control groups. The treatment success rates at the conclusion of the sclerotherapy session were 85.0% for the study group and 83.3% for the control group (P = 0.829), indicated that the difference was not statistically significant.

Table 2 Comparison of the clinical effects 1 month after surgery between the two patient groups.

Clinical effects	Study group (n = 80)	Control group (n = 78)	χ2	P value	
Cured	68 (85.0%)	65 (83.3%)	0.376	0.829	
Improved	11 (13.8%)	11 (14.1%)			
Invalid	1 (1.2%)	2 (2.6%)			

The comparison of the clinical effects between the study group and the control group was presented in Table 2. One month after endoscopic sclerotherapy, the effective rates were 98.8% for the study group and 98.0% for the control group, with no significant difference observed (P = 0.829). Table 3 illustrates the comparison of postoperative complications. In the control group, 11 patients reported postoperative anal pain, while only three patients in the study group experienced this complication, indicating a significant difference (P = 0.022). Additionally, anal swelling was reported in four patients from the study group and 12 patients from the control group, which was also statistically significant (P = 0.031). Neither group experienced urinary retention, perianal infection, ectopic embolism, or anal edema, and no significant differences were found for these complications (P > 0.05).

Table 3 Complication rates one month after surgery in both groups.

Complication	Study group (n = 80)	Control group (n = 78)	χ2	P value	
Anal pain	3 (3.8%)	11 (14.1%)	5.208	0.022	
Anal swelling	4 (5.0%)	12 (17.9%)	4.680	0.031	
Urinary retention	0	0	–	–	
Perianal infection	0	0	–	–	
Ectopic embolization	0	0	–	–	
Anal edema	0	0	–	–	

The comparison of the two treatments was shown in Table 4. The cost of surgery and length of stay in study group were significantly less than those in control group (all P < 0.05). The dosage of polidocanol were similar, and there was no significant difference (P > 0.05).

Table 4 Comparison of relevant indicators for treatment between the two patient groups.

Indicators	Study group (n = 80)	Control group (n = 78)	t	P value	MD
(95% CI)	
The cost of surgery (RMB)	3,846.58 ± 1,890.82	3,969.08 ± 2,331.66	−0.363	0.717	[−788.82 to 543.82]	
Length of stay (d)	4.73 ± 1.65	4.67 ± 1.66	0.173	0.863	[−0.47 to 0.57]	
The dosage of polidocanol (ml)	1.90 ± 0.13	1.88 ± 0.15	0.883	0.379	[−0.02 to 0.06]	

Discussion

Hemorrhoids are one of the most common anorectal diseases that can occur at any age and of any sex and have been reported to occur in half of the population over 50 years (Altomare & Giuratrabocchetta, 2013). Despite the relatively good results reported in the literature, the significant haemorrhoidectomy complications include postoperative pain, urinary incontinence, stricture, and bleeding. Thus, dramatic changes have occurred in treating hemorrhoids in the past few decades, including non-surgical modalities such as sclerotherapy. Sclerotherapy is widely used in Western countries to cause fibrous reactions in the submucosal and haemorrhoid tissues. Sclerotherapy has gradually been recognized as a minimally invasive treatment for internal hemorrhoids in adults.

In 1928, Blanjold first used sclerotherapy for hemorrhoids, but it was gradually replaced due to severe complications. Subsequently, sclerotherapy has gradually become a well-established clinical treatment. Usually, sclerosing agents are critical to efficacy. Different sclerosants have their own advantages and disadvantages and are not identical in treatment. Aluminum potassium sulfate and tannic acid is obviously more useful than phenol in almond oil for injection sclerotherapy; ALTA was more effective than PAO in hemostatic (Yano & Yano, 2015). In a study comparing 3% polidocanol and 5% phenol, polidocanol needs less treatment frequency and higher patient satisfaction (Nastasa et al., 2015). Three percent polidocanol foam vs 3% polidocanol liquid, foam polidocanol is more effective and equally safe compared to liquid polidocanol (Moser et al., 2013). PAO is effective for internal hemorrhoids up to grade III, while ALTA has shown efcacy in treating prolapsing in internal hemorrhoids at grades II, III, and IV (Yamana, 2017). Polidocanol has the advantages of being safer and more effective, and it can effectively reduce complications and improve patient satisfaction (Figueiredo et al., 2022).

Polidocanol foam hardener belongs to the surface activator, which has the advantages over other hardening agents (Lobascio et al., 2021): First, the foam inside it naturally replaces the blood in the blood vessels, and the degree of dilution to the minimum so that the drug concentration in the vein to get a better grasp. As the volume of the bubble becomes smaller, the surface area of the hardening agent increases the area in contact with the inner membrane, which improves the therapeutic effect and can significantly reduce the total dose injected. Second, the foam sclerosing agent can be evenly distributed on the surface of the vascular endothelium, which extends the contact time between the sclerosing agent and the endothelium. Finally, the strong cohesion of foam means that it can be aspirated and reinjected after the first injection. 3% polidocanol foam sclerotherapy has already become an effective conservative treatment for symptomatic second- and third-degree hemorrhoids (He & Chen, 2023). In a comparative study on the treatment of grade I–III hemorrhoidal disease, the efficacy of polidocanol foam sclerotherapy was found to be superior to rubber band ligation (Yamana, 2017).

However, the endoscopic polidocanol sclerotherapy of internal hemorrhoids also has postoperative complications. Related articles show that postoperative complications of sclerotherapy of internal hemorrhoids are anal swelling and anal pain, et al., and they are almost curable (Jacobs, 2014). Although the complications are curable, they reduce the patient’s quality of life. Medical technology has made significant progress, but patients’ demand for minimally invasive technology and an excellent medical experience remains unchanged. Currently, most hospitals use air foam hardening agent to treat internal hemorrhoids. However, our study found that polidocanol carbon dioxide foam sclerosant can be effective in reducing the occurrence of postoperative complications. This may be due to the different half-lives of air and carbon dioxide when combined with polidocanol, which results in varying degrees of foam stability. The combination of air and polidocanol has the longest half-life, leading to a prolonged contact time of the sclerosing agent with the tissue, thereby causing longer-lasting irritation post-surgery. Additionally, air is not easily absorbed by the human body, further contributing to increased inflammation. Carbon dioxide, as a common physiological gas, is frequently used as the foundational gas for establishing pneumoperitoneum in laparoscopic surgery and is readily absorbed by the body, thereby reducing the occurrence of complications. This innovative method of medication deserves further promotion.

As this study is retrospective and involves a relatively small sample size, it may impact the accuracy, reliability, and generalizability of the research findings. To minimize potential errors, we accurately recorded cases through the hospital’s electronic medical record system. Therefore, although retrospective cohort studies have inherent limitations, such as recall bias, the use of objective medical records helps mitigate the distortion of results caused by such biases. Moreover, we enhanced the validity of the study by strictly assessing and recording complication scores through patient follow-up visits at the hospital.

The limitations of this study are that it is a single-centre study and the sample size needed to be larger. The study still needs a multicenter, large sample, prospective randomized controlled clinical trial for further confirmation. We lack in-depth research on the mechanisms by which carbon dioxide foam can reduce complications, and in the future, we will further explore its underlying mechanisms.

In conclusion, the clinical efficacy of endoscopic polidocanol carbon dioxide foam and polidocanol air foam is comparable. Sclerotherapy of endoscopic polidocanol carbon dioxide foam for internal hemorrhoids can effectively reduce the occurrence of anal swelling and anal pain.

Supplemental Information

Supplemental Information 1 Dataset.

Additional Information and Declarations

Competing Interests

The authors declare that they have no competing interests.

Author Contributions

Jiahui Xiang performed the experiments, analyzed the data, prepared figures and/or tables, authored or reviewed drafts of the article, and approved the final draft.

Shichao Li performed the experiments, analyzed the data, prepared figures and/or tables, authored or reviewed drafts of the article, and approved the final draft.

Tengjiang Yu performed the experiments, prepared figures and/or tables, and approved the final draft.

Qingfeng Jiang performed the experiments, prepared figures and/or tables, and approved the final draft.

Xia Jiang performed the experiments, prepared figures and/or tables, and approved the final draft.

Yong Lan conceived and designed the experiments, performed the experiments, authored or reviewed drafts of the article, and approved the final draft.

Human Ethics

The following information was supplied relating to ethical approvals (i.e., approving body and any reference numbers):

The protocol was approved by the Medical Ethics Committee of the Affiliated Traditional Chinese Medicine Hospital to Southwest Medical University (approval number: BY2023033).

Data Availability

The following information was supplied regarding data availability:

The raw measurements are available in the Supplemental File.

Clinical Trial Registration

The following information was supplied regarding Clinical Trial registration:

This experiment is a retrospective clinical study and is harmless to humans.

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
