# Peer review of "The effect of endoscopic polidocanol carbon dioxide foam for internal hemorrhoids: a retrospective study"

_PeerJ, doi:10.7717/peerj.20252_

## Round 0.1 · original submission · Major Revisions

· Academic Editor

Major Revisions

Please respond to the comments from the reviewers.

Reviewer 1 ·

Basic reporting

Language and Clarity:
The manuscript contains several grammatical errors and unclear phrasings that could hinder readability. For example:

In the Methods section: "Among them, endoscopic polidocanol sclerotherapy was administered 78 cases were treated..." should be rephrased for clarity.
Consider revising sentences such as "Both groups of the clinical effects, surgical costs, hospitalization time, and the dosage of polidocanol were the basically same similar..." for clearer communication.

Suggested Improvement:
It is recommended that the manuscript undergo professional English language editing to improve clarity and readability for an international audience.

Literature References:
The manuscript references relevant literature but lacks discussion of recent advancements in sclerotherapy.

Suggested Improvement:
Add more recent references and provide a broader discussion of current trends in sclerotherapy and hemorrhoid treatment.

Experimental design

The research question is clear and relevant. However, the rationale for using a retrospective design could be better explained.
Suggested Improvement:
Include a power analysis or discuss the study's limitations due to sample size. Consider clarifying why the statistical significance for some outcomes is minimal.

Validity of the findings

The statistical analysis is appropriate, but the power of the study should be discussed, especially given the small sample size.

·

Basic reporting

1- The manuscript uses professional language; however, there are several grammatical errors and awkward phrasing that impact readability. For example:
• The basically same similar (Results section).
• The texture of blood vessels is clear (Methods section).
I recommend thorough editing by a professional language editor such as Grammarly or native English-speaking colleague.
2- The introduction is clear and provides sufficient context for the study’s relevance. However, a more comprehensive review of existing studies on carbon dioxide foam’s application in sclerotherapy—particularly its physiological benefits—would strengthen the background.
• Expand the discussion on prior findings related to foam stability and absorption rates between air and carbon dioxide.
• Address any knowledge gaps in previous studies that this research aims to fill.
3- The tables are well-structured and clearly labeled. However, statistical outputs, such as confidence intervals or effect sizes, should be included to complement p-values.
4- The clinical relevance of Figures 3 and 4 is not explicitly clear. Add captions that better link the images to the study’s objectives and outcomes.

Experimental design

5- While the study addresses a clinically relevant question, its retrospective design introduces potential biases. The authors acknowledge the need for future multicenter, randomized trials, but it is essential to explicitly discuss how biases (e.g., selection bias, confounding) were mitigated in the current study.
6- Explain why one month was chosen as the follow-up period. Would longer-term outcomes provide more meaningful insights into recurrence rates or delayed complications?

Validity of the findings

7- While the study’s focus on carbon dioxide foam is noteworthy, similar findings have been reported in studies on varicose veins and other sclerotherapy applications. The novelty of the current study should be more explicitly stated.
8- The significant reduction in postoperative anal swelling and pain with carbon dioxide foam is a key finding. However, the discussion lacks sufficient exploration of the underlying mechanisms. For example, why does carbon dioxide’s absorption reduce these complications?
9- Another limitation not explicitly mentioned is the potential variability in surgical techniques, despite the claim that all surgeries were performed by attending physicians.

Additional comments

Nothing

---

## Round 0.2 · Minor Revisions

· Academic Editor

Minor Revisions

Dear Author,

While the reviewers are satisfied with the manuscript, there are still several instances that require your attention. Firstly, please clarify the abstract by clearly describing outcome variables, emphasizing statistical results and effect sizes where applicable. Secondly, thoroughly review the scientific writing, particularly in the Methods section; for example, many instances where "will be" is used should be corrected to the the past tense. These issues are common when English is not your first language. Therefore, I strongly recommend revising your manuscript thoroughly with the assistance of an expert in scientific writing.

**Language Note:** The Academic Editor has identified that the English language must be improved. PeerJ can provide language editing services - please contact us at [email protected] for pricing (be sure to provide your manuscript number and title). Alternatively, you should make your own arrangements to improve the language quality and provide details in your response letter. – PeerJ Staff

Reviewer 1 ·

Basic reporting

Authors has been responded to all the comments and now this article can be published

Experimental design

ok

Validity of the findings

ok

Additional comments

No

·

Basic reporting

The author addressed my remarks

Experimental design

The author addressed my remarks

Validity of the findings

The author addressed my remarks

Additional comments

The author addressed my remarks

---

## Round 0.3 · Minor Revisions

· Academic Editor

Minor Revisions

Dear Author,

Thank you very much for your meticulous revision. However, I noticed that the referencing in your manuscript does not follow the PeerJ-recommended format before I accept the article.

Best regards,

---

## Round 0.4 · accepted · Accept

· Academic Editor

Accept

Thank you for the revisions you have made to your manuscript, which is now ready for publication.